# Pathology of Diabetes-Induced Immune Dysfunction

**DOI:** 10.3390/ijms25137105

**Published:** 2024-06-28

**Authors:** Michael Alexander, Eric Cho, Eiger Gliozheni, Yusuf Salem, Joshua Cheung, Hirohito Ichii

**Affiliations:** Division of Transplantation, Department of Surgery, University of California, Irvine, CA 92868, USA; michaela@hs.uci.edu (M.A.); ejcho5@hs.uci.edu (E.C.); egliozhe@uci.edu (E.G.); ymsalem@uci.edu (Y.S.); joshukc2@uci.edu (J.C.)

**Keywords:** diabetes, immune dysfunction, immune senescence, infection, medication side effects, vaccine uptake, latent autoimmune diabetes in adults

## Abstract

Diabetes is associated with numerous comorbidities, one of which is increased vulnerability to infections. This review will focus on how diabetes mellitus (DM) affects the immune system and its various components, leading to the impaired proliferation of immune cells and the induction of senescence. We will explore how the pathology of diabetes-induced immune dysfunction may have similarities to the pathways of “inflammaging”, a persistent low-grade inflammation common in the elderly. Inflammaging may increase the likelihood of conditions such as rheumatoid arthritis (RA) and periodontitis at a younger age. Diabetes affects bone marrow composition and cellular senescence, and in combination with advanced age also affects lymphopoiesis by increasing myeloid differentiation and reducing lymphoid differentiation. Consequently, this leads to a reduced immune system response in both the innate and adaptive phases, resulting in higher infection rates, reduced vaccine response, and increased immune cells’ senescence in diabetics. We will also explore how some diabetes drugs induce immune senescence despite their benefits on glycemic control.

## 1. Introduction

Type 2 diabetes (T2DM) is a chronic illness normally occurring in adults, especially the elderly, characterized by an impaired response to insulin (insulin resistance) leading to impaired glucose control and thus chronic hyperglycemia. In the latter stage of T2DM, it leads to the death of the beta-cells in the pancreas that produce insulin, a phenomenon named beta-cell exhaustion [1].

A major side effect of diabetes is the increasing susceptibility to infections. In upper respiratory viral infections, pre-existing diabetes in COVID-19 patients has been shown to be both a comorbidity that leads to a more severe infection [2] and also a predictor of those patients developing long COVID-19 [3]. Similar impairment in diabetic patients with higher glycated hemoglobin (HbA1c) was also shown to occur in response to the original severe acute respiratory syndrome (SARS) [4] as well as influenza [5] and pneumonia [6].

One common comorbidity is diabetic foot infection, which commonly results from diabetic foot ulcers (DFUs) [7]. DFUs are open sores found at the bottom of the feet of individuals with diabetes, and around half of all DFUs become infected. Diabetic patients often do not even notice the wound or infection and thus do not seek treatment in time to prevent or treat the infection [8]. The two most common types of infections are bacterial and fungal infections. Common types of bacterial infections include cellulitis, folliculitis, and erythrasma while the most common types of fungal skin infections include tinea pedea and cutaneous mycoses. Otitis externa, necrotic infections, and mucormycosis are especially concerning infections that continue to be common despite antibiotic use and better blood glucose control [9]. Diabetic skin infections can become even more severe if not treated quickly enough due to the chance of superinfections [7]. Superinfections are a second infection that spreads before the first infection is completely treated. Diabetic skin infections are also difficult to treat due to the lack of efficacy from systemic antibiotics. The lack of efficacy results from both the poor delivery of antibiotics to the infection as well as increasing antibiotic resistance [10]. Due to the rising incidence of diabetes mellitus, diabetic skin infections will not only increase the healthcare burden but will also exacerbate the antibiotic resistance crisis.

Urinary tract infections (UTIs) are not only common but more severe and difficult to treat in patients with diabetes mellitus. Urinary tract infections are the most common type of infection that occurs in individuals with type 2 diabetes [11]. Diabetic UTIs are more common in women, and women who have been diagnosed with diabetes for at least 6 months have higher rates of UTIs than women who were recently diagnosed with diabetes [12]. Another risk factor for UTIs is age, and the possibility of a UTI increases as diabetic patients age. The most common bacteria that cause UTIs in patients with diabetes are *Escherichia coli*, *Klebsiella pneumoniae*, *Pseudomonas auregonosa*, *Enterobacter pneumoniae*, *Proteus* spp., *Proteus* spp., and enterococci [12,13]. Diabetic UTIs also have fungal causes, and *Candida* spp. is the most pervasive [14]. Diabetic UTIs are more likely to be caused by antibiotic-resistant pathogens such as fluoroquinolone-resistant uropathogens and vancomycin-resistant enterococci [12]. Specifically, the gram-negative bacteria display a concerning amount of resistance to common antibiotics such as trimethoprim-sulphamethoxazole, cephalosporins, amikacin, and amoxicillin/clavulanic acid [13]. Diabetic UTIs are also a risk factor for longer hospitalization and serious complications like bacteremia, azotemia, and septic shock. Mortality, morbidity, and relapse are all higher in diabetic individuals [12]. Without better solutions for treating UTIs in diabetic patients, both medical costs and bacterial resistance will continue to rise.

The various difficult comorbidities of diabetes mentioned above make it important to understand how diabetes leads to immune dysfunction. Chronic diabetes, especially those with less well-controlled blood glucose, seems to induce a similar pattern of immune dysfunction as those found in the elderly. While T1DM has also been associated with the accelerated aging phenotype in the immune system, likely because of chronic inflammation, there is a lack of studies that may explain this phenomenon [15]. The accelerated immune aging in T1DM seems to predominantly happen in the first 30 years of life, thus limiting comparison to immune senescence that is normally expected in the elderly [15]. As such, this review will focus on T2DM because of the higher availability of studies on how it leads to immune senescence. In this review, we will examine the various pathologies that can explain this immune dysfunction and how they can explain the rise or worsening of diabetes comorbidities.

## 2. Diabetes-Induced Inflammaging May Explain Diabetes-Related Comorbidities

A critical feature of aging is chronic low-grade inflammation, termed “inflammaging” [16]. Numerous elderly individuals develop inflammaging as they age, which causes an increase in disability, morbidity, accelerated aging, and death [17]. Interestingly, type 2 diabetes itself gives rise to the same inflammaging activities [16]. Inflammaging seems to explain how diabetes cascades onto later comorbidities such as beta-cell dysfunction that worsens T2DM, rheumatoid arthritis, and periodontitis. In this section, we will describe the pathogenesis of inflammaging from T2DM, how inflammaging leads to different types of disorders, and the efficacy of treatments.

### 2.1. How Does T2DM Lead to Inflammaging?

T2DM is a chronic metabolic disorder characterized by insulin resistance, β-cell dysfunction, and hyperglycemia, and is known to increase reactive oxygen species (ROS) [18,19]. Importantly, ROS and hyperglycemia from T2DM have been implicated as risk factors leading to the inflammaging process. In this section, we will look at how ROS are increased in the conditions of T2DM and how they lead to inflammaging.

ROS are the source of inflammaging caused by the antioxidants and hyperglycemia of T2DM [20]. ROS include oxygen-free radicals such as superoxide anion (O_2_^−^), hydroxyl (·OH), hydrogen peroxide (H_2_O_2_), hypochlorous acid (HOCl), hydroperoxyl (HO_2_), singlet oxygen (O_2_), perhydroxyl radicals (HO_2_), and alkoxy (RO) [21]. Moderate or low levels of ROS bring benefits such as the maturation of cells, cellular apoptosis, and the progression of cellular cycles [22]. However, in hyperglycemic conditions of T2DM, the activation of the polyol pathway, hexosamine, and Protein Kinase C and the formation of advanced glycation end products and glycolysis intermediates arise, leading to the overproduction of ROS and the induction of oxidative stress. [23]. In contrast to moderate or low levels of ROS, high levels of ROS in T2DM conditions are known to damage the cells, such as proteins, DNA, lipids, and cellular functions [24]. To add on, according to a study from Eugene and colleagues, oxidative stress biomarkers were notably higher in patients with T2DM in contrast to non-diabetic patients. This result gives evidence of T2DM inducing oxidative stress [25].

With the overproduction of ROS and oxidative stress in T2DM, the promotion of transcription factors such as activator protein-1 (AP-1) and nuclear factor-kappa B (Nf-Kb) produce pro-inflammatory cytokines [26], leading to an inflammatory stage [27]. As stated earlier, T2DM, a chronic disease, induces low-grade chronic inflammation, contributing to and accelerating “inflammaging” (Figure 1). Research from Irene and colleagues provided a cross-sectional study on 122 T2DM patients and 54 non-diabetic patients to analyze inflammation levels in T2DM patients. This study showed that the accumulation of low-density lipoprotein (LDL) and poor glycemic control resulted in a significantly higher inflammatory biomarker in T2DM patients compared to non-diabetic patients [28]. In the following sections, we will review how inflammaging can explain the rise in diabetes comorbidities (Figure 1).

### 2.2. T2DM-Derived Inflammaging Leads to Beta-Cell Death

Pancreatic β-cells produce the hormone insulin, which has a pivotal role in glucose homeostasis [29]. Interestingly, under the condition of inflammaging from T2DM, beta-cell damage occurred. An in vitro study of non-obese diabetic (NOD) mice by Stephens and colleagues found that the tumor necrosis factor-α (TNF-α) pathway may have been the inflammaging-associated mediator that induced the apoptosis of islet cells [30]. In follow-up studies, therapeutic targets regarding reducing TNF-α levels induced by inflammaging in T2DM were evaluated. Treatment by vitamin D3 and chromium picolinate (CrPic) supplementations, proven to reduce TNF-α levels, have been examined and have been shown to prevent β-cell damage and increase insulin sensitivity in T2DM patients [31]. Adding on, the therapeutic method using a caspase inhibitor has also been studied, since caspase activation was found to be a crucial factor for the TNF-α induced apoptosis of beta-cells. Importantly, a study by Stephens and colleagues found that caspase inhibitors protect the isolated beta-cells from NOD mice [30].

### 2.3. T2DM-Derived Inflammaging Leads to the Development of Rheumatoid Arthritis

Rheumatoid arthritis (RA) is a symmetrical, chronic autoimmune inflammatory disease characterized by inflammatory arthritis that is known to increase in risk with T2DM [32,33]. Bone erosion and distortion from cartilage destruction of the joints and the weakening of tendons and ligaments are features of RA. This painful disease eventually starts causing cardiovascular disease and damage to not only the bones but skin, eyes, kidneys, and lungs [34]. The exact reasoning behind why RA occurs from T2DM is not exactly given, but there are growing data that inflammaging from T2DM may be causing the progression of RA. A study by Ming-Chi and colleagues, a study regarding the association between low-grade chronic inflammation in T2DM and RA, was conducted. The study demonstrated that patients with T2DM had elevated risks of RA from inflammation by T2DM [32]. The reasoning behind why patients with T2DM have an increased risk of RA has not yet been identified, but recent case-control studies have shown evidence of an affiliation between inflammation from T2DM and RA. Therefore, further research should be constructed for reasoning and therapeutic methods to cure RA in patients with inflammaging from T2DM.

With the hypothesis of inflammaging from T2DM occurring RA, one of the treatments to be studied is anti-interleukin-1 (IL-1). A study by Piero and colleagues investigated if treatment with anti-IL-1 can improve the inflammatory parameters of patients with both T2DM and RA [35]. The treatment with anakinra, an interleukin-1 receptor antagonist, was given to patients with both T2DM and RA and was shown to decrease the percentage of glycated hemoglobin (HbA1c%) [35]. As stated earlier, RA induces a higher chance of cardiovascular disease and is one of the leading mortality causes of RA [36]. Interestingly, a 1% decrease in HbA1c% was shown to decrease 15% of the occurrence of cardiovascular disease from RA [37].

### 2.4. T2DM-Derived Inflammaging Leads to the Development of Periodontitis

Periodontitis is a serious disorder caused by inflammation from infection in the gum and increases in risk with aging [17]. If periodontitis is left untreated, the bone structure supporting the teeth will be destroyed [38]. T2DM is a known risk factor for gingivitis [39] and periodontitis [40,41], with the pathophysiology remaining underexplained. There is still a lack of evidence that this finding is associated with other types of diabetes, such as gestational or T1DM [41]. Patients with T2DM can develop gingivitis without pre-existing dental plaque normally responsible for the pathology [39]. The severity of the hyperglycemia, not the DM diagnosis itself, affects the periodontium tissue [42] and its vulnerability to developing periodontitis [39] (Figure 1).

Inflammaging may explain how hyperglycemia in T2DM leads to increased periodontitis. Recently, a study with streptozotocin-induced mice models from Qian and colleagues found that diabetes source periodontitis through glucose transporter (GLUT-1)-driven macrophage inflammaging. The study was conducted with diabetic mice (18-week-old) models, and the results showed inflammatory bone loss. Interestingly, the same study conducted with aged (20-month-old) mice showed a worse degree of inflammatory bone loss due to periodontitis. This evidences that diabetic models had accelerated inflammaging and caused more severe degrees of periodontal damage [43]. To support the idea of inflammaging causing periodontitis, a recent study with streptozotocin-induced mice models from Peng and colleagues found that inflammaging accelerated the gingival senescence from the NLR family CARD domain containing 4 protein (NLRC4) phosphorylation [44]. Gingiva is commonly stated as gum and is the primary host defense of periodontium [45]. As a result, diabetic mice with inflammaging effects were found to have an increase in p16 markers, which represent a greater degree of periodontitis, giving evidence of inflammaging causing periodontitis [44].

As more data have been proving that inflammaging causes periodontal damage, therapeutic mechanisms have also been talked about. The study by Qian and colleagues observed the degree of periodontitis levels with the p16 marker and revealed that the treatment of metformin significantly lowered the p16 levels compared to the control group with no treatment of metformin [43]. The study from Peng and colleagues, examining the degree of gingival tissue damage observed with p16 and p21 expressions, established that the treatment of metformin weakened the senescence of gingival tissue in inflammaging mice models. For both studies, a decrease in p16 and p21 levels represented less periodontal damage [43]. In conclusion, this suggests how inflammaging is a strong pathway that increases vulnerability to gingivitis and periodontitis in T2DM patients, linked to how well glycemic control is achieved in these patients. Interestingly, metformin that is already in use as standard treatment may have additional benefits on inflammaging in addition to its effects on glycemic control.

## 3. Diabetes Increases Senescence of Immune Cells from the Bone Marrow

### 3.1. How Diabetes Induces Senescence in the Bone Marrow

Hyperglycemia in diabetic patients induces the senescence of immune cells in the bone marrow through several pathways (Figure 2).

First, hyperglycemia induces a pro-inflammatory state by expanding the bone marrow adipose tissue (BMAT) population. Hyperglycemia induces β-galactosidase activity and the adipogenic differentiation of bone marrow stem cells (adipogenic genes *aP2*, *Lpl*, and *Pparγ*) and increases lipid accumulation in the bone marrow [46] while reducing their osteogenic differentiation potential [47]. Uncontrolled diabetes with Hba1C >7% increases BMA compared to well-controlled diabetes patients with HbA1C <7% [48]. This expansion of bone marrow adiposity (BMA) has been consistently shown to increase pro-inflammatory responses and maintain a steady inflammatory state [49]. This pro-inflammatory state is similar to that found in the aging population, where 10–20% of bone marrow stem cells (BMSCs) are expressing SASP [50] and inducing a decrease in the progenitor cell population and their stem cell differentiation ability [51].

The second cause of the reduction in progenitor cells and their differentiation ability in the bone marrow is likely caused by increased reactive oxygen species (ROS) by BMAT [52,53]. This is further confirmed by the response to ROS scavengers, which significantly reduced ROS levels in BMAT [54].

A third cause is how hyperglycemia induces the senescence of bone marrow mesenchymal stem cells [55] by causing an increase in MSC autophagy. This response to hyperglycemia seems to be a secondary effect of the increase in oxidative stress. This was confirmed by the ability of N-acetylcysteine (antioxidant) and diphenyleneiodonium (DPI, inhibitor of NADPH oxidase) equally to block autophagy and prevent senescence [55].

The hyperglycemia-induced increase in the BMA population causes a significant increase in cytokines upregulated in aging [56]. Senescent cells are characterized by the expression of the senescence-associated secretory phenotype (SASP). These secretagogues include interleukin-1α (IL-1α), interleukin-1β (IL-1β), interleukin-6 (IL-6), nuclear factor Κb (NF-Κb), transforming growth factor β (TGF-β), p21, p16, chemokine ligand 2 (CCL2)/monocyte chemoattractant protein 1 (MCP-1), and chemokine (C-X-C motif) ligand (CXCL1/2) [57], which are themselves associated with local pro-inflammatory responses, especially IL-6 [58]. Using an in vitro co-culture model, BMAT was shown to inhibit B-cell lymphopoiesis at the differentiation stage from progenitor cells to pre–pro B-cells. The progenitor cells instead differentiate toward the myeloid lineage [59]. This is similar to the decline in B-cell lymphopoiesis that occurs in middle age [60] and late stages of life [59,61,62]. BMAT is theorized to inhibit B-cell lymphopoiesis by increasing the population of myeloid-derived suppressor cells (MDSCs) in mononuclear cells (CD11b^+^Ly6C^+^Ly6G^−^) through IL-1 mediated inhibition [63]. A second route is the activation of inflammasomes, such as nod-like receptor 3 (NLRP3) [64], which also negatively affects T-cell proliferation [65] through degeneration in the thymus [66].

CCL2/MCP-1 upregulation that resulted from an increased BMA induces a higher expression of cyclooxygenase-2 (COX-2) in the bone marrow environment [67,68]. COX-2 is metabolized to prostaglandin E2 (PGE2), which promotes the differentiation of dendritic cells toward T-regulatory cells and MDSCs [69,70]. This resulted in the reduction in CD8 T-cells’ response when normally responding to infected cells, as well as the reduction in the antigen-presenting cell population [70].

Restoring glycemic control in diabetes is shown to increase antioxidant enzyme activity and decrease superoxide production, which reduces oxidative stress [47]. A reduction in oxidative stress has been shown to counter BMAT expansion, redirecting the bone marrow cells toward osteogenesis even in older cell populations [71]. In the following section, we will evaluate the effect of diabetes treatments on BMAT expansion.

### 3.2. Effect of Diabetes Treatments on Bone Marrow Adiposity

Treatments of diabetes vary in their effect on bone marrow adiposity and may explain some of the failures in preventing immune senescence in diabetic patients. Metformin is commonly used as a frontline treatment to improve insulin sensitivity. However, metformin increases bone marrow adiposity, prompting osteogenic genes (*RunX2*, *OPN*, and *OCN*) and adipogenic genes (*Ppar-γ*, *Cebpα*, and *Scd1*) in vivo. In contrast, in vitro results showed the opposite effect, where metformin inhibits adipogenesis and promotes osteogenesis [72]. This conflicting result may be explained by considering the balance of BMAT and MSC in filling space in the bone marrow stroma. Further studies showed that metformin induces MSC apoptosis both in vitro and in vivo, explaining the filling of the bone marrow stroma by BMAT [72].

A second class of diabetes drugs is thiazolidinedione (glitazones, TZD) [73]. One of the most common drugs of this class is rosiglitazone. An in vivo mouse femur fracture model showed that rosiglitazone increased BMAT compared to a non-rosiglitazone-treated control. These mice also have a greater bone volume but greater bone porosity. In vitro, the rosiglitazone treatment induced more adipogenesis and reduced osteogenesis [73]. A similar effect is seen with the use of pioglitazone [74,75]. Further characterization of this effect found that Adipsin, a cytokine released by adipocytes, is responsible for the increased BMAT after the rosiglitazone treatment of MSC. The rosiglitazone treatment causes BMAT expansion, during which Adipsin is the most upregulated through the PPARγ acetylation-dependent pathway. An Adipsin knockout mice model showed the inhibition of the BMAT expansion response to rosiglitazone [76].

Gastric bypass surgery is used as a treatment for obese patients and has been found to improve the glycemic control of obese T2DM patients. In a study on gastric bypass patients with or without diabetes, the patients initially showed similar BMAT parameters in both groups. After 6 months post-surgery, diabetic patients showed a decline in BMAT and increased unsaturated lipids. This BMAT decline was not found in the non-diabetic group [77].

In conclusion, the negative effect of metformin and thiazolidinedione on BMAT may explain the immune senescence of T2DM patients despite intensive glycemic control. In obese T2DM patients, this may lead to the consideration of increasing the priority of gastric bypass over pharmacologic options. Alternatively, new PPARγ-sparing TZD analogs, such as MSDC-0602K, offer the benefits of thiazolidinedione without increasing the adipogenesis and senescence of bone marrow cells [74]. In the interim, no data have been reported yet for GLP-1 analogs such as semaglutide or liraglutide, a more advanced treatment for diabetes and the newest entry to the pharmacologic arsenal against obesity, on their effect on immune senescence or on bone marrow adiposity.

## 4. How DM Affects the Innate and Adaptive Immune System

It is observed that several innate defense mechanisms work less than optimally due to DM [78]. Innate immunity is one of the largest tools for fighting primary infections through macrophages. DM leads to a reduced expression of Fcγ receptors on macrophages and monocytes [79]. This leads to a less effective initial response as it causes a decreased rate of endocytosis of a foreign pathogen, thereby slowing subsequent steps in the immune response of DM individuals. Other immune molecules that are affected by hyperglycemic conditions include natural killer (NK) cells, which seem to display problems regarding the surveillance of virally infected cells. DM changes the transcription of the NKp46 receptor (NCR1) and endoplasmic reticulum (ER) stress-induced reduction in NK2GD receptors. In T2DM, ER stress markers such as BiP and PDI mRNAs had a 2.2 times increase. In addition, the Inositol-requiring transmembrane kinase endoribonuclease-1α (IRE1α) pathway of the unfolding protein response (UPR) is increased by 1.98 times in SXBP1 mRNA in diabetics. Collectively, all these variables combined lead to major reductions and or changes to the production and presence of NK2GD receptors on NK cell surfaces. When comparing the mRNA transcription for NCR1 in diabetics to healthy individuals, a significant decrease in mRNA levels was observed [80]. This suggests that hyperglycemic conditions continue to silence the innate response as both receptors play a key role in the cytokine response when surveilling potentially infected cells. When these receptors are not present or present in reduced numbers, there is less likelihood of activating their cytokine response, which includes the release of interferon γ (IFN-γ), TNF-α, IL-10, IL-6, and granulocyte-macrophage colony-stimulating factor (GM-CSF) [81]. By reducing or inhibiting the release of IFN-γ, it handicaps the subsequent adaptive immune system response by not allowing for an increased major histocompatibility complex (MHC-1) presence on neighboring cells and macrophages and not increasing the differentiation of TH1 cells, which will later lead to the activation of CD8^+^ killer T-cells. By reducing IL-6 production, it impairs the humoral response by reducing the differentiation of B-cells into plasma cells, which can then release antibodies into the bloodstream to activate systems such as complement, opsonization, and neutralization. By reducing IL-10 production, the immune system is limited in its ability to dial back the immune response after the infection is eliminated [81]. This leads to increased recovery times and susceptibility to further infection.

The adaptive immune response is also affected by hyperglycemic conditions induced by DM. Research shows that several parts of the adaptive immune system have reduced functionality compared to non-diabetics [82], starting with major changes to the leukocyte recruitment process. Specifically, neutrophils seem to be the most affected group in this process. When comparing diabetics to non-diabetics, there was a significant decrease in CXCL1 and CXCL2 cytokine production [82]. This means there is less recruitment of neutrophils from blood to infected tissue, significantly reducing pathogen elimination and tissue healing. This is further supported by data that show significantly reduced ROS production, overcoagulation, and decreased degranulation in T2DM patients, which are all necessary parts of the initial adaptive immune response when it comes to neutrophils [82,83,84]. By impairing these essential first steps of the adaptive immune system, there is continued stress placed on the rest of the immune system to combat and make up for the lack of pathogen elimination and tissue healing. To make up for this lack of adaptive response, the body releases a hyperinflammatory response. It risks extreme damage to itself to try at all costs to eliminate the pathogen that is infecting it, despite what damage the host may incur. This is caused by the delay of TH1 cell-mediated immunity, which is necessary for pathogen elimination and the development of CD8+ cytotoxic T-cells (CTLs). This reduced ability of the innate immune system to communicate to the adaptive immune system and then the adaptive immune system’s impaired signaling to its other parts explains how detrimental DM can be to the host infection response (Figure 3).

### How DM Affects Vaccine Uptake in the Elderly

Another area of consideration is the reduced vaccine uptake in DM patients. Vaccine uptake requires the interaction between B-cells and naïve T-cells, which also gets impaired in the aging process (Figure 3). As one ages, not only does the rate of lymphopoiesis decrease but the immune system begins to favor myeloid differentiation instead of lymphoid differentiation [85]. This has the direct effect of shifting the immune system from one that is a balance between memory cell lymphocytes and naïve cell lymphocytes toward one that is more concerned with memory cell lymphocytes. Due to this shift, the flexibility that makes the adaptive immune system so effective is lost and thus makes it much more difficult for the elderly to generate novel high-affinity antibodies and T-cell receptors (TCR). Specifically, it has been observed that the reduced IL-7 release and decreased VDJ recombination are the cause of decreased humoral response effectiveness [86,87]. In the cell-mediated immune response, that accumulation of previous infections or very severe infections can cause significant deletions in the TCR diversity [88,89]. In addition to this diversity loss, it has been observed that thymic involution and the hematopoietic progenitor cell decrease are also responsible for the reduced creation of naïve T-cells, which are necessary to generate a large TCR repertoire [90,91]. Given that the elderly struggle to maintain diversity in their B-cell and T-cell receptors, and that they have a reduced production of naïve cells, it should come as no surprise that these effects are increased in diabetics and occur at an even younger age than expected for them [92]. In fact, because of the constant hyperglycemic conditions and high leptin levels of diabetics, they have overly excessive T-cell activation and expansion, which directly causes a reduction in their TCR repertoire and is responsible for many of the comorbidities that are acquired post-DM diagnosis [93].

The real-life statistics show that specifically for hepatitis B, diabetics when vaccinated had a 75.4% protection rate compared to the 82.0% of non-diabetics [94]. In addition, it was also recommended that diabetics should try and receive vaccination at younger ages to confer long-lasting immunity. When compared to non-diabetics at age 60 or older, the diabetics had a seroconversion rate of 58.2% compared to the non-diabetics at 70.2% [94]. Because diabetic patients have further-aged immune systems than non-diabetics, vaccinations before age 40 are recommended to confer better seroconversion rates compared to non-diabetics [95]. In a study on booster vaccines for diabetics, they found that annual re-vaccination was more required for diabetics than non-diabetics because even after receiving the vaccination, their antibody levels decreased faster than for non-diabetics [96]. The duration of T2DM also seems to be a contributing factor to vaccine uptake as it was shown that individuals with T2DM for longer compared to T2DM individuals who had been less affected by T2DM had higher seroconversion rates than long-term T2DM patients [97]. Even T1DM patients had higher seroconversion rates when compared to T2DM, and the higher A1C levels observed in T2DM patients is a major contributing factor to how effective vaccine seroconversion will be [98,99]. In conclusion, T2DM individuals and specially aged T2DM individuals should increase their vaccination frequency and try to control or decrease their A1C levels since both factors strongly affect one’s seroconversion rate and effectiveness in fighting off infection post-vaccination (Figure 3). Receiving vaccination at younger ages will also allow the individuals to take advantage of their more robust immune system that is yet to suffer from having decreased hematopoiesis differentiation and increased myeloid differentiation.

## 5. Latent Autoimmune Diabetes in Adults and Its Immunopathology

### 5.1. Latent Autoimmune Diabetes in Adults as a Heterogeneous Diagnosis

Latent autoimmune diabetes in adults (LADA) is a form of diabetes where patients do not fit into the classical designations of type 1 diabetes (T1DM) or T2DM. The commonly accepted definition of LADA classifies patients with slowly progressive β-cell autoimmunity in which blood glucose levels can be controlled without insulin treatment [100]. While these patients initially appear to be phenotypically similar to T2DM patients, they present with T1DM autoimmunity that manifests with varying levels of activity [101]. Just as expressed in patients with T1DM, LADA patients test positive for glutamic acid decarboxylase (GAD) autoantibodies [102], but unlike patients with T1DM, it has been shown that LADA patients have worse long-term glycemic control compared to T2DM patients [103]. In addition, the presence of *TCF7L2*, a gene commonly predisposing individuals to T2DM, has been found in patients with LADA [104]. There has also been evidence showing that the risk of coronary artery disease, stroke, and cardiovascular mortality is similar in LADA and T2DM patients [105]. Nevertheless, LADA includes a wide umbrella of different manifestations of disease given its broad definition, and it has been shown to have great heterogeneity, with studies distinguishing its wide manifestation of the genetic and immunological factors found in both T1DM and T2DM [106]. Therefore, although there are no specific studies examining LADA and its effect on ROS prevalence, it can be theorized that the similar hyperglycemic environment seen in LADA as well as its sharing of T2DM genetic factors would likewise cause ROS levels to be elevated in patients with LADA. This would entail the possibility of LADA-induced inflammaging as outlined in the previous section. However, when looking into the pathophysiology of LADA, two distinct mechanisms of interest could potentially be affected by or affect inflammaging, which will be further explored.

### 5.2. Type 1 Diabetes-like Pathological Mechanism

In T1DM, one of the main forms of autoimmunity against β-cells comes from CD8^+^ T-cell activation and differentiation into autoreactive cells [107]. As characterized by Mallone et al., many pre–pro-insulin (PPI) epitopes are targeted by autoreactive CD8^+^ T-cells strictly in patients presenting with T1DM, which are present within β-cells, and they developed assays enabling this identification between healthy and T1DM patients [108]. For LADA patients, there have been a few physiological explanations to provide insight as to why the T1DM autoimmunity seen in LADA patients is so delayed. In a study conducted by Sachdeva et al., their group compared the activity of PPI-specific CD8^+^ T-cells derived from LADA and T1DM patients’ blood samples [109]. They found that the quantity and activity of the PPI targeting CD8^+^ T-cells were lower in LADA patients compared to T1DM patients, and most importantly, this difference persisted when comparing older or younger T1DM patients [109]. However, as the authors admit, the fact that the subjects in the LADA group were very recently diagnosed and the heterogeneity of the disease mean that the findings cannot allow for a conclusive mechanistic explanation [109]. There is a real possibility that the PPI CD8^+^ T-cells could eventually reach the levels and activity of T1DM patients over time, and it is also possible for a wide range of unscreened LADA patients to already have this similar PPI CD8^+^ T-cell prevalence and activity. Nonetheless, many questions remain about this mechanistic pathway for LADA patients. It is yet to be seen whether this mild CD8^+^ T-cell activity is present at birth for these patients or if there is some specific trigger for the autoimmune response later in life. Since there is evidence of LADA being independent of body mass index [110], an unknown immunological response could possibly trigger the autoimmune response later in life.

A possible explanation could be the natural inflammaging process. Interestingly, there is evidence that the gut microbiota plays a role in both islet autoimmunity and inflammaging (Figure 4). As shown by Fransen et al., when the gut microbiota were inoculated into young germ-free mice from older mice, they showed increased levels of several types of helper T-cells and an increase in TNF-α [111]. Under these inflammatory conditions, it has been shown that there is an increase in intestinal epithelial cell shedding, causing gaps to form within the lining of the intestine, thus increasing intestinal permeability [112]. Moreover, a study conducted by Bosi et al. has shown an increase in intestinal permeability preceding T1DM pathogenesis in humans [113], and in a study by Costa et al., the pathogenesis of T1DM was induced in mice when streptozotocin was translocated into pancreatic lymph nodes [114]. What this means for patients with LADA is that there is a plausible mechanism for which the natural process of aging, and thus inflammaging, initiates the onset of their autoimmune diabetes. Additional studies should be conducted to characterize whether there is a change in intestinal permeability in LADA patients as well as increased intestinal inflammation. There should also be investigations into whether these factors are influenced by the genetic susceptibility of LADA patients for T1DM.

### 5.3. Alternate Type 2 Diabetes-like Pathological Mechanism

It has been well-established that a risk factor for the manifestation of T2DM is obesity [115] and that the chronic low-grade inflammation associated with obesity drives insulin resistance [116]. As described in the previous section, the TNF-α pathway can directly lead to β-cell damage and death [21]; however, a less understood autoimmune pathway could provide some more insight into the mechanism affecting obese patients with LADA [117]. Tyrosine phosphatase IA-2 (IA-2) autoantibodies have been found to be present in both LADA and T2DM patients [117,118], and it has been associated with an increased incidence of insulin therapy for T2DM patients [118]. In the study conducted by Tiberti et al., it was found that a subset of the IA-2 autoantibody, IA-2_(256–760)_, was present in 30% of patients positive for the GAD autoantibody and 3.4% of patients negative for the GAD antibody and another IA-2 autoantibody, IA-2_IC(605–979)_, in a population of LADA patients [117]. The IA-2_(256–760)_ autoantibody was shown to be the only antibody with a positive correlation with BMI in T2DM patients in another study by Buzzetti et al. [119]. Additionally, only these strictly IA-2_(256–760)_ autoantibody-positive T2DM patients phenotypically resembled patients who are obese with T2DM and have a longer time without the need for insulin treatment over 7 years compared to GAD autoantibody-positive patients [119]. With all this in mind, it is possible to suggest that the low-grade inflammation generated in obese individuals could trigger this alternate autoimmune pathway that is non-classical to the T1DM autoimmunity seen in T1DM and LADA patients. Considering the variety of genotypes and phenotypes that fall under the umbrella of LADA, it seems plausible that there could be an additional autoimmune pathway that can lead to LADA that is phenotypically similar to T2DM. These findings highlight the need for further investigation into autoimmunity in T2DM. Furthermore, from what was discussed, there are very complex pathological mechanisms in LADA that have the potential to interact with each other and affect inflammaging. Likewise, it could be speculated that inflammaging itself could be a factor in triggering autoimmunity in LADA patients, which could then feedback into triggering more inflammaging due to the metabolic environment generated by diabetes.

## 6. Implication for Treatment Choice for Diabetes

To solve the immune dysfunction that arises from diabetes, one common target is to treat the diabetes-induced inflammation response. Inflammation is caused by the greater oxidative stress burden that arises in a high glycemic state and is a common cause of diabetes comorbidities [120]. However, several diabetes drugs that have beneficial effects on oxidative stress [120], such as metformin and thiazolidinedione, unexpectedly may induce immune senescence in T2DM patients [72,74]. Other classes of pharmaceutical approaches to diabetes [120] thus need to be considered. Several diabetes drugs that have been shown to have beneficial effects on the Nrf2 oxidative pathway are liraglutide [121,122], sitagliptin [123], and dihydromyricetin [124,125]. The effects of these drugs on immune senescence have not been characterized.

In obese patients, a gastric bypass was shown to avoid the BMAT increase [77] that induces immune senescence. Based on markers of oxidative stress, namely glutathione, superoxide dismutase, and catalase, 8-oxo-7,8-dihydro-2’-deoxyguanosine (8-oxodG) and 8-oxo-7,8-dihydroguanosine (8-oxoGuo), the data suggest that a gastric bypass is beneficial in reducing oxidative stress [126] and thus may need higher prioritization in obese T2DM patients.

## 7. Conclusions

Diabetes, especially T2DM, alters the immune response in a similar manner to that found in the aging process. A similar phenomenon has also been reported on T1DM patients; however, this has not been as well characterized. The current theory for this pathology is the process of inflammaging, a chronic low-grade inflammation. Inflammaging, along with other mechanisms, increases the senescence of bone marrow cells including those that are involved in the immune response lineage. Interestingly, several drugs used to treat diabetes, including metformin and thiazolidinedione, also induce similar senescence responses. This side effect should thus be considered by patients and doctors when selecting their diabetes treatment. The combined effect of the impaired immune system and side effect of some diabetes drugs leads to increased vulnerability to infections, reduced vaccine uptake, and damage to the β-cells, and may explain the atypical phenomenon of LADA.

## Figures and Tables

**Figure 1 ijms-25-07105-f001:**
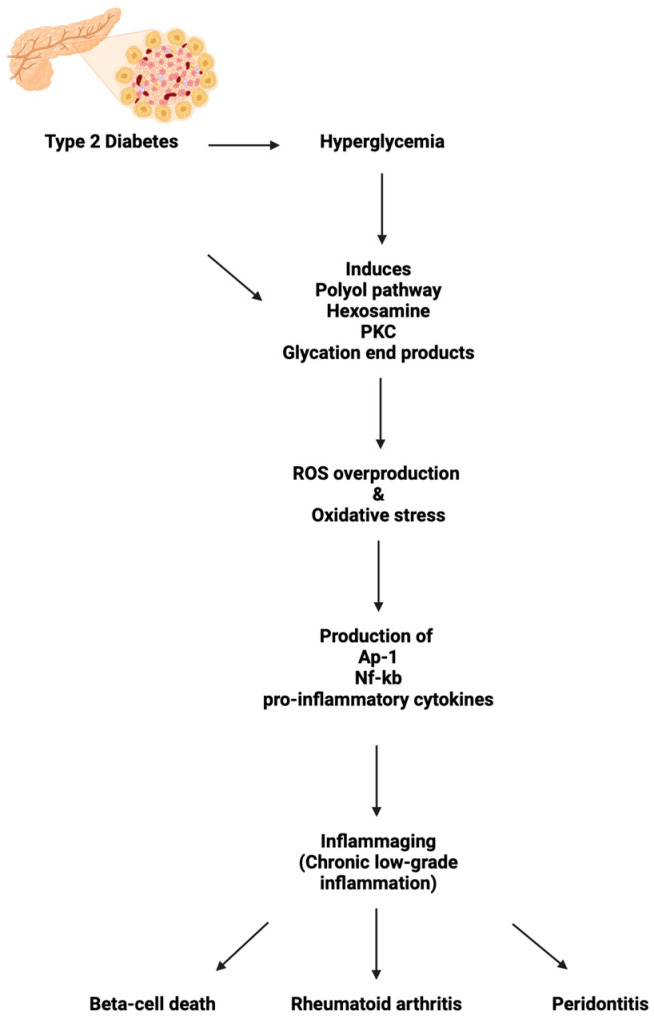
Diabetes-induced inflammaging pathway.

**Figure 2 ijms-25-07105-f002:**
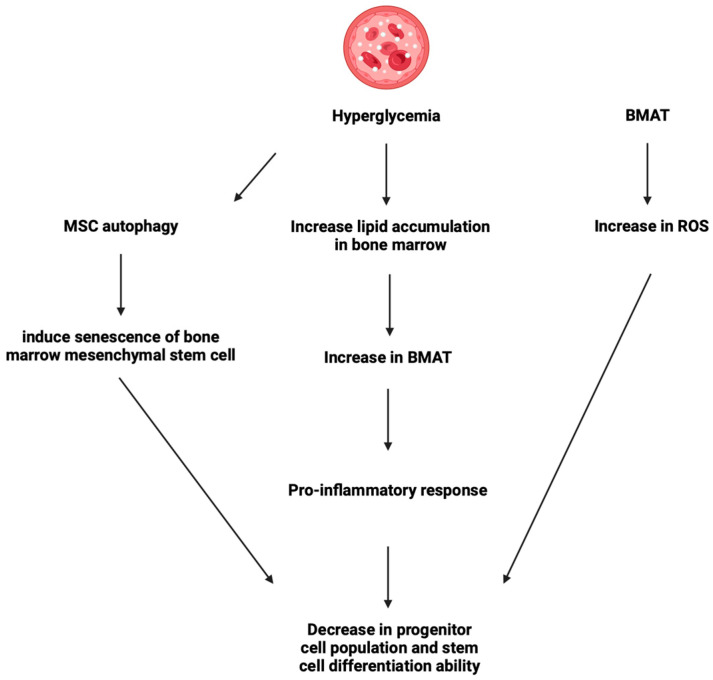
Various pathways of how hyperglycemia leads to senescence of immune cells in the bone marrow.

**Figure 3 ijms-25-07105-f003:**
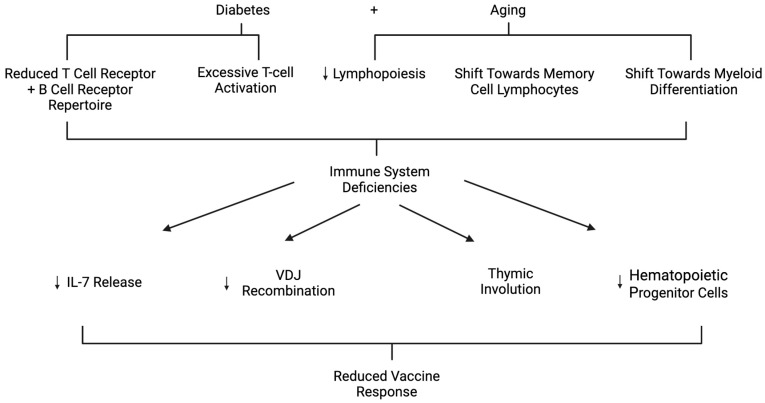
The pathways on how diabetes and aging both reduce response following vaccination. Arrows represent reduction.

**Figure 4 ijms-25-07105-f004:**
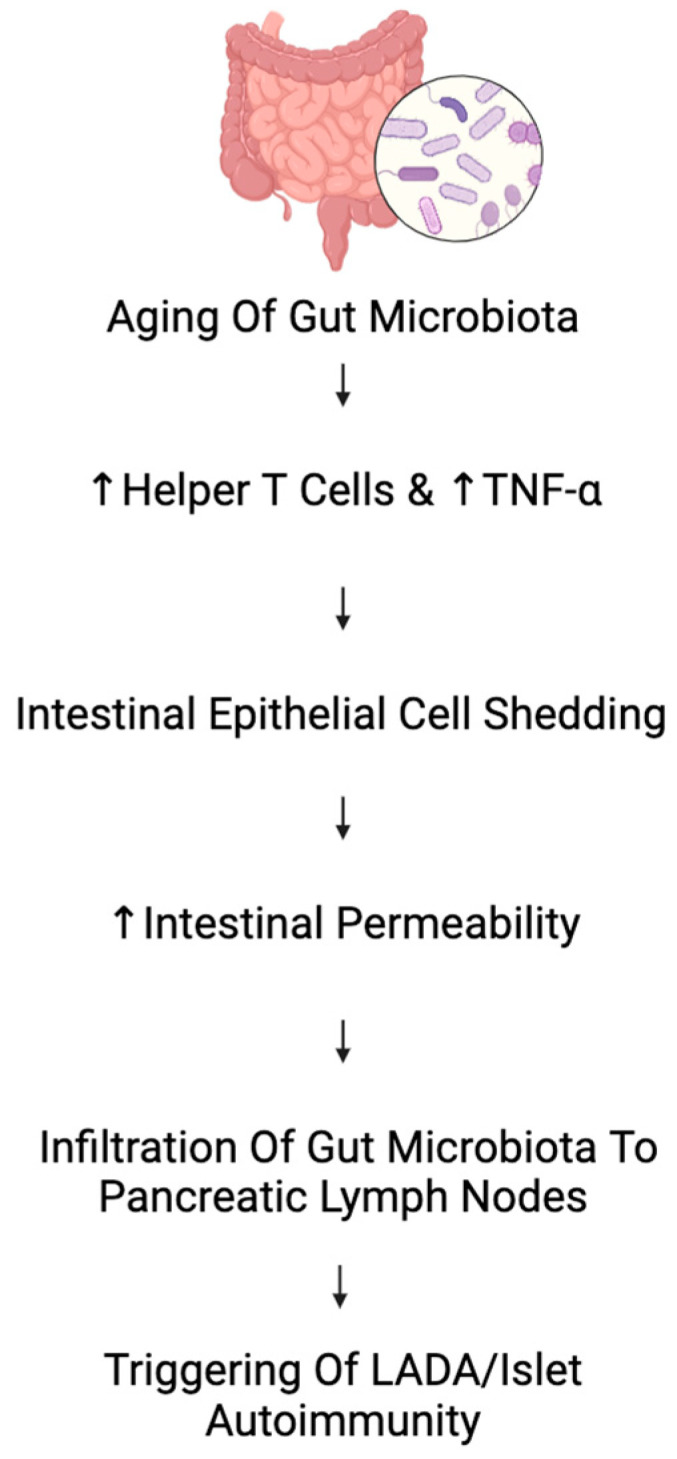
The gut theory of latent autoimmune diabetes in adult (LADA) origin. Arrow represent “Increase”.

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
