# Peer review of "Pathology of Diabetes-Induced Immune Dysfunction"

_ijms, 2024, doi:10.3390/ijms25137105_

Round 1

Reviewer 1 Report

Comments and Suggestions for Authors

This is an interesting review that outlines the various ways in which diabetes is able impair the strength of immune response. I only have  a few comments for the authors:

1. I think the abstract needs to contain a short summary of the authors findings with a conclusion as it provides little detail at the moment.

2. The review seems to concentrate on type II diabetes. The authors should provide some explanation as to why there is a paucity of information on type I diabetes.

3. In view of the inflammation associated with diabetes I`m surprised that there is no information on oxidative stress or suggested treatments to mitigate the damage from free radical induced oxidation in this common disorder.

Author Response

This is an interesting review that outlines the various ways in which diabetes is able impair the strength of immune response. I only have a few comments for the authors:
1.    I think the abstract needs to contain a short summary of the authors findings with a conclusion as it provides little detail at the moment.
Thank you for this suggestion, we modified the abstract to expand and better explain the findings and conclusions.
2.    The review seems to concentrate on type II diabetes. The authors should provide some explanation as to why there is a paucity of information on type I diabetes.
While T1DM has also been associated with accelerated aging phenotype in the immune system, there’s lack of studies addressing this phenomenon. One study suggests that the accelerated immune aging in T1DM predominantly happens in the first 30 years of life, and that chronic inflammation is the main cause of immune aging. We’ve added this commentary in the manuscript.
3.    In view of the inflammation associated with diabetes I`m surprised that there is no information on oxidative stress or suggested treatments to mitigate the damage from free radical induced oxidation in this common disorder.
We added a new section summarizing suggested treatments for T2DM that avoids the issue of immune senescence, while having positive benefits on diabetes-induced oxidative stress.

Reviewer 2 Report

Comments and Suggestions for Authors

In manuscript #IJMS-3009857 "Pathology of diabetes-induced immune dysfunction" by Alexander et al, the authors reviewed age-related changes in the diabetic immune response. It is scientifically sound and contains sufficient interest and originality to merit publication. We do not have any criticism of the explanations in the references, but more figures and tables would make it more accessible to the reader.

Author Response

In manuscript #IJMS-3009857 "Pathology of diabetes-induced immune dysfunction" by Alexander et al, the authors reviewed age-related changes in the diabetic immune response. It is scientifically sound and contains sufficient interest and originality to merit publication. We do not have any criticism of the explanations in the references, but more figures and tables would make it more accessible to the reader.
    Thank you for your review and suggestion, we’ve added new figures to summarize and better explain the sections for the reader.